# Influenza A Hemagglutinin Passage Bias Sites and Host Specificity Mutations

**DOI:** 10.3390/cells8090958

**Published:** 2019-08-22

**Authors:** Raphael T. C. Lee, Hsiao-Han Chang, Colin A. Russell, Marc Lipsitch, Sebastian Maurer-Stroh

**Affiliations:** 1Bioinformatics Institute, Agency for Science Technology and Research, Singapore 138671, Singapore; 2Department of Epidemiology, Center for Communicable Disease Dynamics, Harvard TH Chan School of Public Health, Boston, MA 02115, USA; 3Department of Medical Microbiology, Academic Medical Center, University of Amsterdam, Amsterdam, 1105 AZ, The Netherlands; 4Department of Biological Sciences, National University of Singapore, Singapore 117558, Singapore; 5National Public Health Laboratory, National Centre for Infectious Diseases, Ministry of Health, Singapore 308442, Singapore

**Keywords:** influenza, passage, host specificity, mutations, adaptation, pandemic, hemagglutinin

## Abstract

Animal studies aimed at understanding influenza virus mutations that change host specificity to adapt to replication in mammalian hosts are necessarily limited in sample numbers due to high cost and safety requirements. As a safe, higher-throughput alternative, we explore the possibility of using readily available passage bias data obtained mostly from seasonal H1 and H3 influenza strains that were differentially grown in mammalian (MDCK) and avian cells (eggs). Using a statistical approach over 80,000 influenza hemagglutinin sequences with passage information, we found that passage bias sites are most commonly found in three regions: (i) the globular head domain around the receptor binding site, (ii) the region that undergoes pH-dependent structural changes and (iii) the unstructured *N*-terminal region harbouring the signal peptide. Passage bias sites were consistent among different passage cell types as well as between influenza A subtypes. We also find epistatic interactions of site pairs supporting the notion of host-specific dependency of mutations on virus genomic background. The sites identified from our large-scale sequence analysis substantially overlap with known host adaptation sites in the WHO H5N1 genetic changes inventory suggesting information from passage bias can provide candidate sites for host specificity changes to aid in risk assessment for emerging strains.

## 1. Introduction

Influenza pandemics typically occur when an influenza virus from animals infects humans and evolves the capacity for human-to-human transmission [1,2,3,4,5]. The viral surface protein hemagglutinin (HA) is critical for recognizing the respective host cell receptors [6]. Previous studies in ferrets have shown that as few as 3–4 HA mutations (along with a mutation in the influenza virus polymerase complex) can be sufficient to enable a highly pathogenic A/H5N1 avian influenza virus to become mammal-to-mammal transmissible [7,8]. Such experiments are controversial [9] and cannot be performed on a wide variety of strains due to cost and ethical issues; indeed, such experiments produce results which cannot be extrapolated across relatively small changes in the genetic background [10]. Alternative data sources that could complement our understanding of genetic sites important for host specificity could provide valuable information about mutations associated with virus adaptation to new hosts. Commonly observed in laboratory practice, influenza viruses mutate at specific positions when cultured or passaged in different cell types (e.g. from different species). These passage bias mutations are assumed to increase fitness of the virus in the respective species and are sometimes necessary to grow in culture at all. Although passage bias mutations occur in almost all studies involving influenza virus culture, only a few studies have investigated passage bias directly and were limited to specific contexts such as effects on inferring phylogenies for one subtype [11,12]. Since passage annotation data is available for tens of thousands of viruses, we wanted to investigate systematically over common subtypes which positions in HA show passage bias and if this information could be linked to host specificity mutations which could also aid in pandemic risk assessment in the future [13,14,15,16,17].

## 2. Materials and Methods

### 2.1. Sequence and Passage Data Sources and Analysis

Passage annotation is available in GISAID’s EpiFlu database, the most complete source for influenza sequences, but annotation submission has historically been heterogeneous without controlled vocabulary [18]. For example, ‘M1’, ‘MDCK3’ or ‘MDCK 2 + 2’ all refer to passage in Madin-Darby Canine Kidney (MDCK) cells. Based on frequency of occurrence and relevance for interpretation in our host specificity context, we classified available ambiguous passage annotations into four broad categories: (i) embryonated hen’s eggs (denoted as “EGG”), (ii) Madin-Darby canine kidney cells (“MDCK”), (iii) α-2,6-sialyltransferase enriched MDCK cells (denoted as “SIAT”) and (iv) original clinical samples (denoted as “ORI”). Table 1 shows the number of classified passage categories for HA sequences from different subtypes (about 80,000 in this analysis).

Full length HA protein sequences from GISAID with passage information that can be classified in the four groups above were aligned with MAFFT [19] and odds ratios for different culture categories were calculated for every pair of amino acids occurring at the same position. Only amino acids with at least 20 occurrences at the respective position were analysed to avoid low frequency biases. Positions with at least one relevant odds ratio of ≥2 or ≤ 1/2 were considered candidate passage bias sites for further analysis. To account for the fact that different subtypes might be preferentially grown in different cell types we calculated the odds ratios only within each subtype. Furthermore, we verified that use of different passage systems in these data does not fluctuate significantly over time to avoid temporally biased sequence variants appearing as passage sites.

### 2.2. Same Strain Different Passage (SSDP) Analysis

Some viral strains are passaged in more than one cell type and represent prime candidates to study passage mutations in context of the same strain genomic backbone. For each of the viral subtypes, we searched for isolates with the same strain name that has been passaged in both EGG cells and mammalian cells (MDCK/SIAT or ORI for human viruses). We counted the amino acid positions that are different between the pair of differentially passaged isolates with the same isolate name. For some strains, there can be several isolates that are passaged in egg cells and several isolates that are passaged in mammalian cells. For such cases, we consider all possible combinations of Egg-mammalian pairs. To reduce bias caused by over counting a single position for these combinations, mutation at a site is counted only once for any given isolate with the same strain name. Hence, the final count of mutations at a given site represents the number of different strains where we found a mutation at the site when differentially passaged in egg and mammalian cells. We excluded H5 data because there were too few examples and this allows to test if analysis without H5 still allows to find host specificity sites also relevant for H5. 

### 2.3. Pair Analysis

In addition to understanding passage bias for single sites, we examined if the association between sites is affected by passage. We tested the association between all the pairs of sites, and considered three pairs of passages, MDCK vs. ORI, MDCK vs. EGG, and ORI vs. EGG, by the following conditional logistic regression method conditioned on time:

Logit(p [*AA1* = 1]) ~ *β*_1_ × *AA2* + *β*_2_ × passage + *β*_3_ × *AA2* × passage + strata(time)
(1)
where *AA1* and *AA2* represent two sites that are considered and *AAi* is equal to 1 if the observed amino acid is the same as the most frequent amino acid variant and 0 if it is the same as the second most frequent amino acid in a given site *i*. The analysis was performed in R using the clogit function in the survival package [20]. We only showed the significant results after Bonferroni correction. The unit of time is month. If *β*_3_ is different from 0, it suggests that the association of AA2 with AA1 differs between two passages. The method assumes that the same proportions of different aa combinations (denoted by *P*_00_, *P*_01_, *P*_10_, *P*_11_) were passed into different cell types, but does not assume that the sample sizes were the same across passages or across time. However, it is possible that the violation of the assumption was related to the balance of the sample sizes from different cell types (for example, if the ratio of the sample sizes of EGG to MDCK was particularly high at one time point, it is possibile that one study passed many samples to EGG and these samples might have very different *P*_00_, *P*_01_, *P*_10_, *P*_11_ from the rest of the population) and therefore we excluded the time periods where the sample size ratio between two passages fell farther than two standard deviations away from the mean in order to minimize this possibility. To increase statistical power, we updated the analysis sets for H1 and H3 subsets (Appendix A).

## 3. Results

### 3.1. Structural Overview of Passage Bias Sites in HA Across Different Influenza A Subtypes

Based on a highly permissive threshold of odds ratio >2 in any culture cell type with any virus subtype, we identified 472 candidate passage bias sites in HA which would cover 83% of all HA positions. To understand how to filter and prioritize these sites, we analysed them in more detail regarding independence of genomic background and the structural positions of these substitutions relative to known functional sites, especially those previously associated with host specificity.

Within a given influenza subtype (e.g., pandemic H1, seasonal H1, H3, H5), the sites where passage bias was detected were mostly shared between different passage types (Appendix A). This is not surprising since preference for one amino acid in eggs often means another amino acid at the same site is preferred in MDCK or original samples.

Passage bias sites were also frequently shared between different influenza subtypes (e.g. pandemic H1, seasonal H1, H3, H5). Interestingly, the extent of sharing (Figure 1) appears to mirror the genetic distance of the subtypes relative to each other known from phylogenetic trees. For example, pandemic H1 shares more sites with H5 than H3. Furthermore, this allows consideration of passage sites shared among multiple subtypes as universal consensus to avoid subtype-specific biases such as culture type preference or temporal variations.

Scoring each passage bias site by the number of different subtypes and different passage cell types and calculating the geometric mean of their odds ratios gave a ranking of sites that were dominantly involved in passage bias, independent of subtype (Appendix A). Next, we wanted to compare if the reliably identified sites with geometric mean odds ratio >5 are robust across different collection/time periods and compared sites using the same described analysis for data until May 2013 against those from June 2013–May 2019 and found good agreement with 45 of 52 (87%) sites of the smaller set being shared (Figure 2).

Occasionally, virus strains can be found in the database where the same strain has been passaged in different cell types (Same Strain Different Passage = SSDP). Especially relevant for us is if the different passage was between mammalian (MDCK/SIAT) and avian (EGG) cell types. These examples are of great value for this study as they represent a clean signal of different passage conditions for the very same starting strain or genomic backbone. Considering only H3, H1s and H1p since H5 data were limited, there are 59 sites shared between at least two subtypes (Appendix A). These conserved direct passage sites are then compared against the two time period sets from the indirect passage sites from large-scale statistical analysis and define our proposed set of robust common passage bias sites. 19 passage bias sites are shared between all three data sets and 54 are shared between at least two (Table 2 and Figure 2). 

These 54 common passage bias sites can be found across the HA structure (Figure 2 and Figure 3) but with higher density in the globular head domain. Interestingly, the distribution of dominant passage sites within the globular head domain was not limited to the immediate receptor binding pocket but rather spread around the globular head domain which also is involved in docking to cells and includes antigenic regions consistent with earlier findings [12]. A secondary passage site cluster was detected in a region required to undergo structural changes for pH-dependent fusion [21,22]. A third cluster is not visible in the structure but is located in the N-terminal signal peptide region. This region seems experimentally understudied for influenza [23] but mutations in a signal peptide can in principle alter secretion efficiency and therefore surface expression. All 3 main clusters make mechanistic sense in terms of what we know about importance for influenza HA biology.

Some of the sites with low geometric mean (below 5) also tend to have a lower number of occurrences in the 18 possible subtype and cell combinations, indicating that although these sites met the criteria of being present in at least 2 out of 3 of the analyses, they may not be as dominant across different subtypes or were just not found in the SSDP analysis. On the other hand, some positions such as 183 and 187 with a relatively high geometric mean of 11.88 and 16.16 respectively were not listed in Table 2 because they were not found in the SSDP analysis and they did not meet the (geometric mean >5) criteria for one of the time period analyses.

### 3.2. Epistasis of Site Pairs Differs by Passage Types

In order to investigate if pairs of sites show passage type dependent co-occurrence, we used a conditional logistic regression approach, conditioned on time, to test for association between sites in seasonal human influenza A viruses (H1 and H3) in regard to passage type. We identified a list of site pairs with significantly different association between passages (Table 3 and Figure 3). The majority of these were between ORI and EGG, suggesting that epistasis might differ between mammalian and avian hosts.

### 3.3. Comparing HA Passage Bias with Host Specificity Mutations

#### 3.3.1. Passage Adaptations cannot Easily Predict Directionality of Host Specific Adaptation Mutations of Specific Strains in Ferret-Adaptation Studies

We compared the identified HA passage bias for seven mutations identified in the gain-of-ferret-transmissibility studies on H5N1 viruses by Herfst et al and Imai et al [7,8]. Passage bias data correctly matched direction of avian to mammalian adaptation (as approximated by EGG vs. MDCK preference) for four mutations (N172D, T174A, Q240L and G242S (H1pdm numbering)), not enough passage data was available for another two (H120Y and N238K) and the opposite/wrong adaptation direction was predicted for only one mutations (T333I). It seems H5 mammalian adaptation is not easy to fully predict with passage data as most H5 strains are grown in eggs and few mammalian examples are available and extrapolating from other subtypes can only be a rough approximation (Appendix A). Although directionality of adaptation appeared difficult to predict fully, suggesting a role of the involved structural positions for adaptation without strongly postulating direction may be possible.

#### 3.3.2. Passage Adaptation Sites Overlap with Previously Known H5 Sites That Affect Host Specificity

The purpose of this study was to test if passage data can help in identifying mutations that may be linked to host specificity changes and hence pandemic risk. Therefore, we compared the different derived lists of passage sites for overlap with the compiled list of host specificity sites in the H5 genetic changes inventory [24]. Figure 4 shows that, as expected, H5 data as independent set is too small to give rise to significant odds ratios for most relevant sites while data even from seasonal viruses, because of their quantity, does get substantial overlaps between common passage and known H5 host specificity sites. Finally and supporting the validity of the consensus approach, the most commonly shared 19 mutations from the overlap of three sets (red in Figure 2 structure) gave the highest odds ratio of 11.85 [95% c.i.3.81–36.9] and significant Fisher exact p-value of 0.0003 when comparing common passage sites with the H5 inventory sites (Figure 4).

## 4. Discussion

Amino acid changes due to passage adaptations occur commonly in viruses grown in laboratory culture. They have been shown to enable the virus to switch from growing well in the original host cells to the lab cultured cells [25,26,27]. While multiple factors and genes determine the ability of influenza viruses to cross the host species barrier [16,28] this study focuses on hemagglutinin which is the main surface protein recognizing the host receptors. Additonally, in hemagglutinin, overlap of antigenic regions and passage sites might lead to reduced efficacy of egg-grown vaccines [12,29]. In this study, we show that passage adaptation sites in influenza viruses are strongly associated with sites involved in host specificity shifts identified previously in animal experiments and from human cases. Across different influenza A subtypes, we identified three main passage bias clusters where adaptation most commonly occurs: (i) receptor binding site and the globular head region, (ii) region that undergoes pH-dependent structural changes and (iii) the N-terminal signal peptide region (Figure 2). When the virus switches from its original host environment to the cell culture environment, many things change: the cell surface receptors and its concentration may be different, pH may be different, the intracellular transport and signal peptide guided secretion may differ in efficiency. Hence, it is not surprising that the most common passage bias sites occur at the observed sites, where they can help the virus adapt to the new cellular environment.

Our study analyses a large number of passage data across multiple influenza subtypes and their associated sequences but the statistical power still depends on the sample size. There are limitations to the approach because, although the sites undergoing mutations necessary for host adaptation can be predicted, the exact residues necessary for the virus to adapt to the new host could not be fully predicted (4 out of 7 correct, 1 wrong, 2 without prediction because of too little data). For the latter, understanding the role of the genetic background of the virus may improve prediction capacity as hinted by the epistatic relations among passage sites we also observed. Alternatively, since the mammalian cell type used here was canine kidney cells, while the ultimate target of interest is human respiratory epithelia and the ferret experiments involved still a third mammalian species, the lack of overlap may point to the limitations of generalizing across mammalian species and/or cell types. It will be interesting to confirm the sites as well as epistatic links through analysis on even more data in future as more data becomes available. Another limitation of the current study is the use of only the hemagglutinin protein to examine passage bias. In spite of the fact that HA plays a key role in the life cycle and host adaptation of the influenza virus, mutations in other viral proteins such as in the polymerase complex (PB2, PB1 and PA), as well as PA-X and the HA-NA balance are also known to play a role in host adaptation and the agreement, or not, with passage sites should be subject for future studies. Additionally, it is unclear how other experimental variables such as temperature and infective dose may affect passage mutations as well as the different number of passage steps.

Above limitations aside, the sites identified from large scale sequence analysis here significantly overlap with known host adaptation sites and show that passage bias can provide candidate sites for host specificity changes to aid in risk assessment for emerging strains [16,17] when combined with a tool such as the FluSurver (https://flusurver.bii.a-star.edu.sg) that allows automated mutation annotation of query sequences with phenotypically relevant information. Last but not least, analysis of this readily available and growing data set of sequences with passage adaptation seems to be an interesting cheap and safe approach for identifying sites associated with adaptation to host-shifts.

## Figures and Tables

**Figure 1 cells-08-00958-f001:**
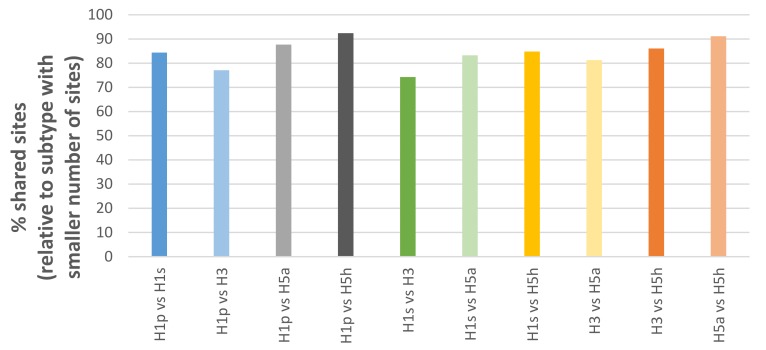
Overlap of passage bias sites between different influenza subtypes.

**Figure 2 cells-08-00958-f002:**
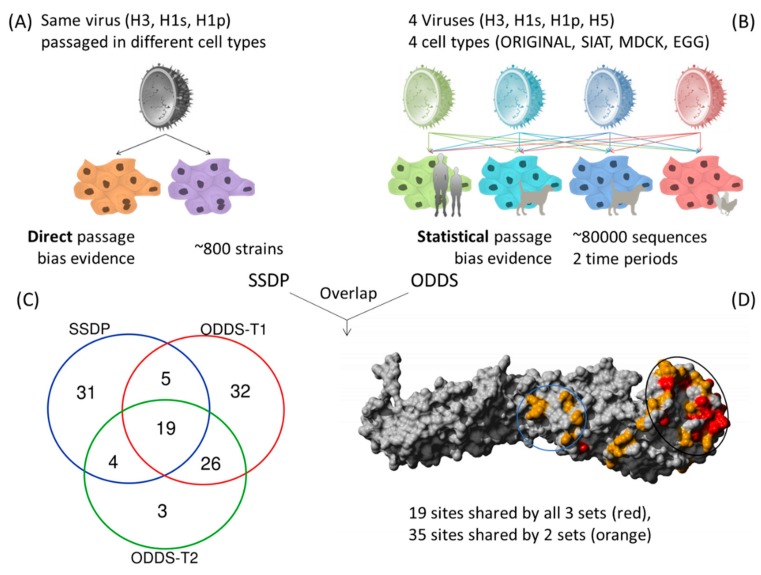
Approach and data types for identification of common passage bias sites. (**A**) Direct evidence from same strain with different passage (SSDP), (**B**) geometric mean of odds ratios >5 from large statistical analysis over two time periods, (**C**) Venn diagram of overlap between SSDP and ODDS sets from two time periods. 54 sites are found in at least 2 of these sets of analyses, (**D**) Structural positions of dominant passage bias sites in HA. Black circle indicates receptor binding head region (i). Blue circle shows region undergoing pH dependent structural changes during fusion (ii). A third identified passage region is not shown as it is in the unstructured signal peptide region (iii).

**Figure 3 cells-08-00958-f003:**
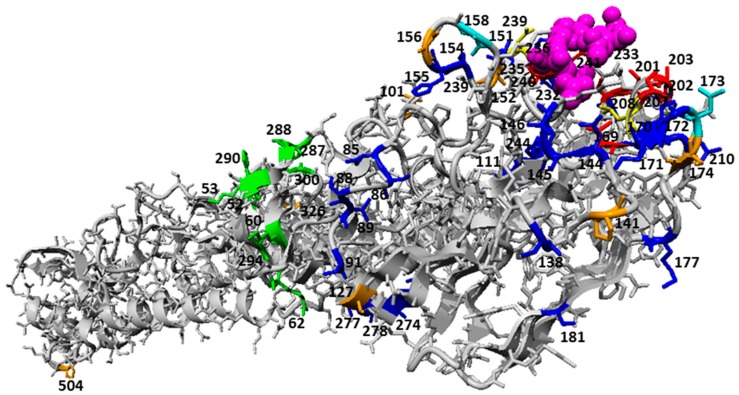
Pairs of sites with significantly different association between passages after Bonferroni correction mapped to HA monomeric structure (PDB 3UBN) in context of common passage bias sites. Structural positions of dominant passage bias sites and sites undergoing epistatic interactions that are under the influence of passage bias adaptations are highlighted in the Figure. The ligands (sialic acid and *N*-acetyl glucosamine) are shown in pink ball representation. Epistatic sites are highlighted in orange. Passage bias sites (with high geometric mean of odds ratio across subtype and cell combinations) in the receptor binding region are colored in red and sites 208 and 239 which overlap with epistatic sites are colored in yellow (i). The rest of the passage bias site in the globular head region are colored in blue. Sites 158 and 173 which overlap with epistatic sites are highlighted in cyan. Sites colored in green lie at a region undergoing pH dependent structural changes during fusion (ii). A third identified passage region is not shown as it is in the unstructured signal peptide region (iii).

**Figure 4 cells-08-00958-f004:**
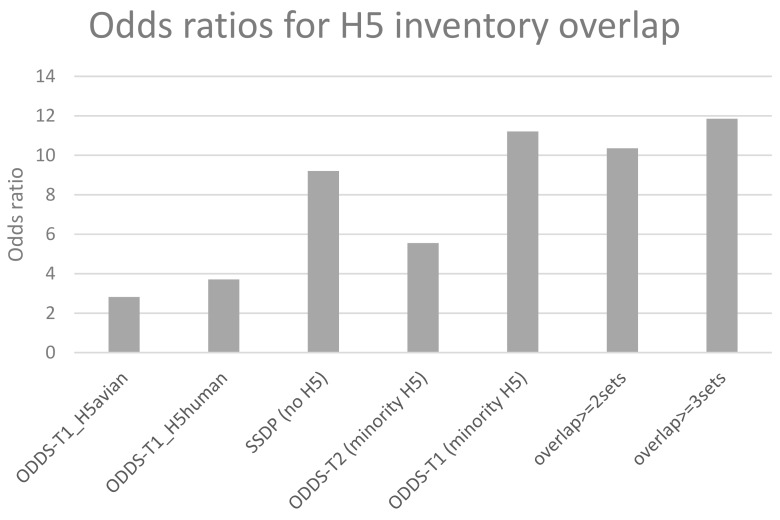
Overlap of different sets of passage sites with host specificity sites in the H5 genetic changes inventory.

**Table 1 cells-08-00958-t001:** Summary of HA sequences used in this study classified into four passage categories.

Subtype	EGG	MDCK	SIAT	ORI	SUM
H1 pandemic (≥2009)	1312	18,059	2103	13,004	34,478
H1 seasonal (<2009)	606	2610	145	342	3703
H3 seasonal	1405	9663	10,423	17,641	39,132
H5 (human)	234	36	0	14	284
H5 (avian)	1439	16	0	186	1641
SUM	4996	30,384	12,671	31,187	79,238

**Table 2 cells-08-00958-t002:** Dominant common passage bias sites in HA. Passage bias sites were identified using (i) same strain different passage (SSDP) analysis, (ii) large-scale analysis of sequence and passage data until May 2013, and (iii) large-scale analysis for sequence and passage data from June 2013 to May 2019. The 54 passage bias sites listed in this table are found in at least 2 out of 3 of the above mentioned analyses. The 19 passage bias sites that are found in all 3 analyses are underlined.

Structural Regions	Sites (H3 Numbering, Starting After Signal Peptide)	Sites (H1pdm Absolute Numbering, Starting at Leading Methionine)	# Occurrence in 18 Possible Subtype + Cell Combinations (4 H1p, 4 H1s, 4 H3, 3 H5 Avian and 3 H5 Human)	Geometric Mean of Odds Ratios from all Subtype/Cell Combinations
Globular Head (RBS)	155 186 187 189 193 194 225 226 227	169 200 201 203 207 208 239 240 241	15 10 14 16 17 13 12 10 13	8.31 6.03 4.11 23.25 11.92 7.59 6.65 16.05 10.73
Globular Head (Around RBS)	131 132 133 137 140 141 144 156 157 158 159 188 196 198 218 221 222 230	144 145 146 151 154 155 158 170 171 172 173 202 210 212 232 235 236 244	16 17 15 13 16 11 18 12 7 15 17 17 14 14 13 11 11 12	11.42 9.25 13.47 10.68 11.92 3.91 81.24 11.11 3.36 12.97 31.17 20.84 8.40 19.90 3.62 3.19 9.97 3.7
Globular Head (Others)	77 80 81 82 101 125 163 167 260 262 263	85 88 89 91 111 138 177 181 274 277 278	12 17 14 15 18 18 14 10 13 16 10	5.9 35.95 6.27 18.62 18.65 24.19 12.41 3.97 5.02 9.43 4.45
pH Structural Change site	45 46 53 54a * 273 275 279 285	52 53 60 62 288 290 294 300	13 14 16 11 14 17 12 14	7.08 9.02 15.26 7.99 12.36 14.31 2.87 11.54
N-terminal (Signal Peptide)	−9 −8 −7 −6 −5 −4 −3 −2	9 10 11 12 13 14 15 16	15 10 16 16 13 15 13 15	12.59 4.92 12.53 10.95 10.44 12.89 12.33 13.36

* 54a (H3 numbering) represents the gap between positions 54 and 55 in H3 viruses. The corresponding position in H1 viruses is position 62 (H1pdm absolute numbering).

**Table 3 cells-08-00958-t003:** Pairs of sites with significantly different association between passages after Bonferroni correction.

Position 1	Position 2	Log Odds Ratio	*P*-Value
**H1N1 ORI vs. EGG in H1 Absolute Numbering** **(H1pdm09 Absolute Numbering in Bracket)**			
13 (13)	101 (101)	1.69	1.11 × 10^−10^
**H3N2 MDCK vs. EGG in H3 Absolute Numbering** **(H1pdm09 Absolute Numbering in Bracket)**			
19 (gap)	176 (174)	1.20	2.22 × 10^−15^
19 (gap)	210 (208)	−1.98	4.17 × 10^−8^
154 (152)	158 (156)	1.31	5.70 × 10^−6^
160 (158)	176 (174)	1.99	0
160 (158)	210 (208)	−3.26	4.81 × 10^−6^
175 (173)	176 (174)	0.90	8.23 × 10^−11^
**H3N2 ORI vs. EGG in H3 absolute numbering** **(H1pdm09 absolute numbering in bracket)**			
19 (gap)	176 (174)	3.00	0
19 (gap)	210 (208)	−1.87	5.75 × 10^−8^
19 (gap)	241 (239)	1.57	2.82 × 10^−7^
19 (gap)	327 (326)	1.55	5.88 × 10^−7^
19 (gap)	505 (504)	1.78	5.69 × 10^−7^
130 (127)	176 (174)	5.85	9.10 × 10^−6^
144 (141)	176 (174)	−0.69	5.79 × 10^−7^
154 (152)	158 (156)	4.15	5.41 × 10^−8^
154 (152)	160 (158)	−1.01	1.32 × 10^−7^
154 (152)	176 (174)	−0.95	4.60 × 10^−8^
154 (152)	327 (326)	−0.99	6.46 × 10^−8^
158 (156)	176 (174)	−0.76	2.02 × 10^−9^
158 (156)	327 (326)	−0.68	4.27 × 10^−7^
160 (158)	176 (174)	4.18	0
160 (158)	210 (208)	−2.85	1.14 × 10^−6^
160 (158)	241 (239)	3.11	1.38 × 10^−6^
175 (173)	176 (174)	2.27	0
175 (173)	210 (208)	−1.23	4.46 × 10^−6^
175 (173)	327 (326)	1.12	7.11 × 10^−6^
175 (173)	505 (504)	1.28	2.13 × 10^−6^

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
