# Peer review of "Influenza A Hemagglutinin Passage Bias Sites and Host Specificity Mutations"

_cells, 2019, doi:10.3390/cells8090958_

Round 1
Reviewer 1 Report
Lee et al identified specific mutations in the influenza A virus (IVA) hemagglutinin (HA) after passages in MDCK cells or chicken eggs. They identified three regions which appeared to be selected after those serial passages. Two of those regions are associated with known functional properties of the HA. The third is in the unstructured N-terminal signal peptide. The sites which they identified overlap with known host adaptations observed in nature. They also found that most frequently biased amino acids were shared between different passages and between different subtypes.
This study is a pure bioinformatics analysis. The authors used existing information from the IAV database for their study. Thus, there is no experimental validation of their findings. The mutations, which they identified were located in known sites and do not represent any novel additional findings.
However, I found it a very novel approach to leverage data in existing databases, to delineate the sequence of selection bias and to compare these changes with those occurring in nature. Such studies will provide additional support for experimental studies and predictions on host IAV adaptation. Because of this, I find this study worth to be published in the Journal Cells.
Minor points.
Please indicate in the abstract that the study has been performed for H1, H3 and H 5. Also, the IAV sequences analyzed were obtained from viruses passaged in a few mammalian and avian cells/tissues. They only considered MDCK cells and embryonated eggs – please be more specific in the abstract.
Line 121. ‘.. types they were found in gave a ranking of sites ..’ There seems to be a word missing.
Table 2: It is not clear to me why a separate numbering has been used for pdmH1N1. If necessary, please explain.
Could the authors provide a hypothesis why the third region, the unstructured and terminal signal peptide, also showed a passage bias?
There is also a potential bias of reporting. Not laboratories have reported their findings. It maybe that only a few major laboratories deposited their data. Could they comment on this aspect in the discussion? Or maybe they addressed this point?
Author Response
Point 1: Lee et al identified specific mutations in the influenza A virus (IVA) hemagglutinin (HA) after passages in MDCK cells or chicken eggs. They identified three regions which appeared to be selected after those serial passages. Two of those regions are associated with known functional properties of the HA. The third is in the unstructured N-terminal signal peptide. The sites which they identified overlap with known host adaptations observed in nature. They also found that most frequently biased amino acids were shared between different passages and between different subtypes.
This study is a pure bioinformatics analysis. The authors used existing information from the IAV database for their study. Thus, there is no experimental validation of their findings. The mutations, which they identified were located in known sites and do not represent any novel additional findings.
However, I found it a very novel approach to leverage data in existing databases, to delineate the sequence of selection bias and to compare these changes with those occurring in nature. Such studies will provide additional support for experimental studies and predictions on host IAV adaptation. Because of this, I find this study worth to be published in the Journal Cells.
Response 1: We thank the reviewer for the supportive comments and have further strengthened the analysis in this revision while maintaining the essence of the approach as described before. In particular, we expanded the comparison to compare two separate time periods to show robustness of the findings as well as an additional subset of cases where the same sample/strain has been passaged with different cell types and show that these orthogonal sets agree well with the known sites and they additionally do identify a small number of new sites that could be important to watch if they occur mutated in future strains.
Minor points.
Point 2: Please indicate in the abstract that the study has been performed for H1, H3 and H 5. Also, the IAV sequences analyzed were obtained from viruses passaged in a few mammalian and avian cells/tissues. They only considered MDCK cells and embryonated eggs – please be more specific in the abstract.
Response 2: Added subtypes and cell types specifically to abstract.
Point 3: Line 121. ‘.. types they were found in gave a ranking of sites ..’ There seems to be a word missing.
Response 3: corrected.
Point 4: Table 2: It is not clear to me why a separate numbering has been used for pdmH1N1. If necessary, please explain.
Response 4: The reason is that the sites are shared between the two subtypes and in order to allow readers to find the respective position in either subtype we provide both numberings.
Point 5: Could the authors provide a hypothesis why the third region, the unstructured and terminal signal peptide, also showed a passage bias?
Response 5: Our hypothesis is as follows (added to paper): “This region seems experimentally understudied for influenza [21] but mutations in a signal peptide can in principle alter secretion efficiency and therefore surface expression. “
Point 6: There is also a potential bias of reporting. Not laboratories have reported their findings. It maybe that only a few major laboratories deposited their data. Could they comment on this aspect in the discussion? Or maybe they addressed this point?
Response 6: It is true that the majority of passage annotations are provided by a small number of reference laboratories who are also the largest volume sequence submitters. However, the relative split between the big labs is similar and data coming from these big labs is derived following very consistent protocols which increases the quality of the data for our analysis. We added the citation to a recent review on history and distribution of passage annotation in different databases [https://www.ncbi.nlm.nih.gov/pubmed/31275610].
Reviewer 2 Report
The study uses public passage bias data through bioinformatics analysis to predict the host specificity mutation in HA. It is interesting, however, it seems like there is no much valuable information which predicts which amino acid mutation will alter the virus fitness or the pandemic risk.
The author mentioned that the previous study showed 3-4 mutations on HA of H5 virus which can confer the ability to transmit between ferrets is not extrapolated. That is true. Because the flu viruses have so many different subtypes and variants. Most of them have particular properties in a specific-strain way. However, performing the passage in the animal is an important way to predict the pandemic risk for public health. Usually, the mutations generated during cell culture cannot match the mutation in the animal.
The transmissibility and virulence of flu viruses are regulated by multiple genes. Although the HA plays a key role in the flu life cycle,HA-NA balance,PA or PA-X also are very important to mediate the characteristic of viruses.
In addition, we do not how exactly the passage performed. Because the temperature, passage times, and infective dose will affect the mutations. I am not sure how important this method is to predict the host adaption.
Author Response
Point 1: The study uses public passage bias data through bioinformatics analysis to predict the host specificity mutation in HA. It is interesting, however, it seems like there is no much valuable information which predicts which amino acid mutation will alter the virus fitness or the pandemic risk.
Response 1: The purpose of this study was to test if passage data can help in identifying mutations that may be linked to host specificity changes and hence pandemic risk. While we show that at the highest level of detail the direction of mutational effects is not captured by this approach, we show that at the level of sites of mutations there is a significant overlap with known host specificity sites. To strengthen our point, in this revision, we expanded the comparison to compare two separate time periods to show robustness of the findings as well as an additional subset of cases where the same sample/strain has been passaged with different cell types and show that these orthogonal sets agree well with the known sites and they additionally do identify a small number of new sites that could be important to watch if they occur mutated in future strains.
Point 2: The author mentioned that the previous study showed 3-4 mutations on HA of H5 virus which can confer the ability to transmit between ferrets is not extrapolated. That is true. Because the flu viruses have so many different subtypes and variants. Most of them have particular properties in a specific-strain way. However, performing the passage in the animal is an important way to predict the pandemic risk for public health. Usually, the mutations generated during cell culture cannot match the mutation in the animal.
Response 2: We agree with the reviewer that extrapolation of transmissibility from animal tests as well as cell culture may be limited by strain-specific effects. In our study, we approach this in two ways. Firstly, we look at which sites are consistently occurring as passage sites across different subtypes (shared between them). These have a higher odds ratio of overlap with known H5 adaptation sites even when only H1 and H3 data are being used. We have added this extra analysis to the revised version. Secondly, we find evidence for the mentioned strain-specific effects within our analysis of epistatic pairs that suggest that passage/adaptation sites are linked to specific other sites away from their structural position.
Point 3: The transmissibility and virulence of flu viruses are regulated by multiple genes. Although the HA plays a key role in the flu life cycle,HA-NA balance,PA or PA-X also are very important to mediate the characteristic of viruses.
Response 3: We very much agree with the reviewer. We have an ongoing project on highly relevant PB2 and other polymerase complex mutations regarding host adaptations including experimental verifications but the results are not ready yet and adding this to the current MS would be out of scope. So, we just added the notion of importance of other genes and factors to the discussion citing a recent review (second sentence of discussion).
Point 4: In addition, we do not how exactly the passage performed. Because the temperature, passage times, and infective dose will affect the mutations. I am not sure how important this method is to predict the host adaption.
Response 4: This is a good point. In a separate smaller unpublished project we had investigated with colleagues how many passage steps are required for the typical passage mutations to occur. The result was that after the second passage most strains have mutated the typical sites already and do not mutate much further with more passage steps. Besides this anecdotal evidence, in this study, due to the large number of sequences with passage adaptations (~80000), our approach is driven by the most common experimental setups that are being used and not sensitive to few outliers. In fact, the majority of passage annotations are provided by a small number of reference laboratories who are also the largest volume sequence submitters. However, data coming from these big labs is derived following very consistent protocols which increases the quality of the data for our analysis. We added the citation to a recent review on history and distribution of passage annotation in different databases [https://www.ncbi.nlm.nih.gov/pubmed/31275610].
Reviewer 3 Report
In this manuscript by Raphael TC Lee et al., the authors explored the possibility of identifying viral mutations associated with host-specificity through the use of sequence and passage data of isolates in the GISAID’s EpiFlu database. They built a set of hemagglutinin sequences, mainly from human viruses (pdm H1N1, seasonal H1N1 before 2009 and seasonal H3N2), and also from H5N1 viruses. The sequences were allocated into four groups, according to whether the sequence is that of the original isolate (ORI) or that of the virus after passage in embryonated egg (EGG), in MDCK cells (MDCK) or in alpha-2,6-sialyltransferase enriched MDCK cells (SIAT). After alignment of the sequences, odds ratios for the different categories were calculated for every pair of amino acids occurring at the same position.
The manuscript is well written. However, parts of the manuscript are somewhat preliminary. There are a number of points that need to be elucidated, and the authors should perhaps try to concentrate on human viruses (pdmH1N1, seasonal H1N1 and seasonal H3N2). As regards H5 viruses, the authors elaborate too much on unsufficient data.
In brief, the authors should try to go deeper in their analysis with human H1 and H3 viruses and omit the part dedicated to H5 viruses. This analysis should be more explicit regarding the dominant residues (precise location in the structure, structural relationships with co-variating residues).
Main findings
Sequence analysis of the sets of HA sequences identified 478 candidate passage bias sites, which cover 84% of all HA positions.The authors thus identified the dominant passage bias sites, mainly in human H1 and H3 viruses. Several of these bias sites were common to several subtypes, with the same position in the structure. A number of passage bias sites are subject to co-variations. Lines 56-64. Did the authors set boundaries for the isolation dates? (e.g. from 2003 to 2018)? Table 1. The first three lines are easy to understand: the HA sequences from human isolates (pdmH1N1, seasonal H1N1 before 2009 and seasonal H3N2) were classified into the four categories. However, the H5 line is unclear and should perhaps be deleted. Are these avian isolates or human isolates (less than 1000 human cases of infection with H5 viruses since 2003)? This is unclear, and could potentially bias the study. Furthermore, as suggested by lines 158-167, this approach seems inadequate for H5 viruses, (ii) because of the low number of MDCK passages available, and (ii) because the EGG vs MDCK preference may not be a proper approximation. Further, for the sake of clarity the table should also include a bottom line with the vertical sums. Lines 82-97. The calculation is unclear. Lines 102-105. Perhaps the authors should try to graphically represent all the HA residues and their odds ratio. For instance with x-scale from 1 to 566 and y-scale representing a rational combination of odds ratio. Considering the human viruses, such a rational combination could be the geometric mean of the two ratios [Ori vs Egg] and [MDCK vs Egg]. Such a representation should allow one to immediately point to the most important residues. Lines 110-111 and Suppl. Table 2. If passage bias site were frequently shared between subtypes, then this could be indicated in suppl. Table 2, perhaps as an eighth column that could somehow combine the odds ratios observed, or as 2-3 supplementary columns with the most relevant types of odds ratios (for human isolates, I would bet on Egg/MDCK and Egg/Ori). Figure 1. As indicated by the title of the y-axis, and as suggested by the graphical values, the graph is redundant (e.g. H1p vs H1s is the same as H1s vs H1p). The graph should only contain 6 bars, and it becomes more obvious that the percentage of shared sites is <70% for all the comparisons involving H3 (lines 111-12). Table 2 and suppl. Table2. Table 2 is confusing. I would propose a table with at least 5 columns: column 1=cluster 1, then cluster 2 then cluster 3 // column 2= the sites in H3 HA1 numbering (grouped according to the cluster) // column 3=the corresponding sites in H1N1pdm absolute numbering // column 4= occurrence in subtype+cell combination // column 5= some type of relevant odds ratio (or a rational combination of relevant odds ratios). Therefore, residues -8, -6 and -2 would be consecutive, either as the top lines or bottom lines of the table, corresponding to the third cluster (signal peptide). Figure 2 and its legend. The legend should indicate that what is shown is a HA monomer, either as a space-filling model or as a ribbon model. Which HA is it? (H1, H3, H5?). The residue numbers of the dominant (or most relevant) passage bias should be clearly indicated (notably those from tables 2 and 3), by choosing a “universal numbering” that could be defined in suppl. table 2. Table 3. The data are interesting. However, the authors should try to find alternative and/or complementary modes of representation, including graphical representation. A number of residues seem to be associated in clusters, both in the MDCK vs EGG and ORI vs EGG for the H3N2 viruses, e.g. [19, 160, 175-176, 210], [154, 158]…, and residue 176 seems to play a prominent role, given the number of times it shows covariations with other residues (with very low P-values). It would probably be useful to locate these associated residues in the HA structure (notably the [19, 160, 175-176, 210] cluster). It is striking that the prominent residues in the co-variation analysis do not appear in the dominant passage bias sites (Table 2). Could the authors propose hypotheses to explain that? Section 3.3 (lines 156-187). As stated above (comments to Table 1), the approach may be inadequate for the H5 viruses, essentially because of the low number of MDCK passages. The authors are elaborating too much on unsufficient data. Because of the very low fraction of H5 viruses with MDCK passages (47 sequences for a total of 1596 H5 sequences), one may question the statistical validity of the whole analysis on H5 viruses. Lines 193-95. The authors have not really shown that “passage adaptation sites … are strongly associated with … host specificity”. Or at least this endeavour was not really convincing for H5 viruses. Perhaps should they instead try to find corroborating data with H1 or H3 viruses. The discussion could try to establish useful correlations and to go deeper in the structural relationships between the dominant residues. Ref 22 is incomplete.Author Response
Point 1: The manuscript is well written. However, parts of the manuscript are somewhat preliminary. There are a number of points that need to be elucidated, and the authors should perhaps try to concentrate on human viruses (pdmH1N1, seasonal H1N1 and seasonal H3N2). As regards H5 viruses, the authors elaborate too much on unsufficient data. In brief, the authors should try to go deeper in their analysis with human H1 and H3 viruses and omit the part dedicated to H5 viruses.
Response 1: We thank the reviewer for this comment and agree that the H5 data itself is insufficient. We have made substantial changes to the analysis looking at subtypes separately and new combinations of the datasets without H5 with a new emphasis on showing that also when excluding H5 data we can use passage data of seasonal viruses (H1p, H1s, H3) to identify known host specificity sites and get significant overlap with the WHO H5 genetic inventory list (H5inv).
Point 2: This analysis should be more explicit regarding the dominant residues (precise location in the structure, structural relationships with co-variating residues).
Response 2: Good point with the co-varying residues in the structure. We now highlight the identified sites in the structure in a new Figure with explicit labels including the co-varying residues from the pair analysis.
Point 3: Lines 56-64. Did the authors set boundaries for the isolation dates? (e.g. from 2003 to 2018)?
Response 3: That was a good question and we noticed that the set of the previous version of the manuscript was limited to data until May 2013 only. To strengthen our analysis, in this revision, we expanded the comparison to include all latest available data until May 2019 and use it to compare two separate time periods to show robustness of the findings.
Point 4: Table 1. The first three lines are easy to understand: the HA sequences from human isolates (pdmH1N1, seasonal H1N1 before 2009 and seasonal H3N2) were classified into the four categories. However, the H5 line is unclear and should perhaps be deleted. Are these avian isolates or human isolates (less than 1000 human cases of infection with H5 since 2003)? This is unclear, and could potentially bias the study.
Response 4: We thank the reviewer for the comment and we agree that H5 viruses from different source organisms can have a different effect on passage bias. For the detailed analysis on all possible subtype and cell combinations and their correlation with H5inv, H5 viruses were split into avian and human datasets for analysis. The breakdown for this number can now be seen in Table 1. We have also made the necessary changes to the methods section to reflect this.
Point 5: Furthermore, as suggested by lines 158-167, this approach seems inadequate for H5 viruses, (ii) because of the low number of MDCK passages available, and (ii) because the EGG vs MDCK preference may not be a proper approximation.
Response 5: We agree with the reviewer, and as we have pointed out before by lines 158-167, that the low number of MDCK passages for H5 viruses is problematic if we were to use it to interpret the direction of host adaptations. As mentioned earlier, we have made substantial changes to the analysis looking at subtypes separately and new combinations of the datasets without H5 with a new emphasis on showing that also when excluding H5 data we can use passage data of seasonal viruses (H1p, H1s, H3) to identify known host specificity sites and get significant overlap with the WHO H5 genetic inventory list (H5inv).
The reviewer raised the possibility that EGG vs MDCK preference may not be a good approximation for avian to mammalian host adaptations. It is true that some other cell types (such as human lung cells) could provide a better approximation. However, because of the lack of data for other cell types, EGG vs MDCK becomes the best cell-based approximation to look at host adaptations with enough statistical power and we show that with the significant overlap with the WHO H5 genetic inventory list even without considering H5 data explicitly.
Point 6: Further, for the sake of clarity the table should also include a bottom line with the vertical sums
Response 6: We thank the reviewer for the suggestion. We now also added the vertical sums to the table.
Point 7: Lines 82-97. The calculation is unclear.
Response 7: We have now added more details to describe how the calculation for the conditional logistic regression is done.
“The analysis was performed in R using the clogit function in the survival package (https://CRAN.R-project.org/package=survival.). We only showed the significant results after Bonferroni correction. The unit of time is month. If β3 is different from 0, it suggests that the association of AA2 with AA1 differs between two passages. The method assumes that the same proportions of different aa combinations (denoted by P00, P01, P10, P11) were passed into different cell types, but does not assume that the sample sizes were the same across passages or across time. However, it is possible that the violation of the assumption was related to the balance of the sample sizes from different cell types (for example, if the ratio of the sample sizes of EGG to MDCK was particularly high at one time point, it is possibile that one study passed many samples to EGG and these samples might have very different P00, P01, P10, P11 from the rest of the population), and therefore we excluded the time periods where the sample size ratio between two passages fell farther than two standard deviations away from the mean in order to minimize this possibility. To increase statistical power, we updated the analysis sets for H1 and H3 subsets (Suppl. Table 2).”
Point 8: Lines 102-105. Perhaps the authors should try to graphically represent all the HA residues and their odds ratio. For instance with x-scale from 1 to 566 and y-scale representing a rational combination of odds ratio. Considering the human viruses, such a rational combination could be the geometric mean of two ratios [Ori vs Egg] and [MDCK vs Egg]. Such a representation should allow one to immediately point to the most important residues.
Response 8: We thank the reviewer for this nice and helpful suggestion. We have now included this as a new plot in the supplementary. However, because the odds ratio for each pair of amino acid residues found at every position is calculated for each passage type against all other passages, we modified the reviewer’s suggestion and show log odds ratio for the top 5% positions in EGG and MDCK passage types for the human viruses. We did plot for both time periods and the plots show reasonably good consistency with each other, and the most important residues could be seen from the plots.
Point 9: Lines 110-111 and Suppl. Table 2. If passage bias site were frequently shared between subtypes, then this could be indicated in suppl. Table 2, perhaps as an eighth column that could somehow combine the odds ratios observed, or as 2-3 supplementary columns with the most relevant types of odds ratios (for human isolates, I would bet on Egg/MDCK and Egg/Ori).
Response 9: We thank the reviewer for the suggestion. Since the odds ratio for each pair of amino acid residues found at every position is calculated for each passage type against all other passage categories, we modified the reviewer’s suggestion and appended to supplementary Table2, an additional column of geometric mean of odds ratio from all subtypes and passage categories. The table legend for Supplementary Table2 has been changed to reflect this. “HA numbering conversion table for influenza subtypes H1, H3 and H5. The last column is the geometric mean of Odds Ratio for all the influenza A subtypes and passage categories combinations examined. ”
Point 10: Figure 1. As indicated by the title of the y-axis, and as suggested by the graphical values, the graph is redundant (e.g. H1p vs H1s is the same as H1s vs H1p). The graph should only contain 6 bars, and it becomes more obvious that the percentage of shared sites is <70% for all the comparisons involving H3 (lines 111-12).
Response 10. We thank the reviewer for pointing this out. We removed the redundant bars and added a new split for the distinction between human and avian H5.
Point 11: Table 2 and suppl. Table2. Table 2 is confusing. I would propose a table with at least 5 columns: column 1=cluster 1, then cluster 2 then cluster 3 // column 2= the sites in H3 HA1 numbering (grouped according to the cluster) // column 3=the corresponding sites in H1N1pdm absolute numbering // column 4= occurrence in subtype+cell combination // column 5= some type of relevant odds ratio (or a rational combination of relevant odds ratios). Therefore, residues -8, -6 and -2 would be consecutive, either as the top lines or bottom lines of the table, corresponding to the third cluster (signal peptide).
Response 11: We thank the reviewer for the excellent suggestion. We agree that grouping the passage sites by structural regions/clusters can give readers a better understanding. We created a new Table 2 following the reviewer’s suggestion with a new table legend.
Point 12: Figure 2 and its legend. The legend should indicate that what is shown is a HA monomer, either as a space-filling model or as a ribbon model. Which HA is it? (H1, H3, H5?). The residue numbers of the dominant (or most relevant) passage bias should be clearly indicated (notably those from tables 2 and 3), by choosing a universal numbering” that could be defined in suppl. Table 2.
Response 12: We agree with the reviewer that a universal numbering should be used. The Passage bias sites and the epistatic sites should be clearly labelled in the H1pdm09 structure (3ubn). We have now made this new figure 3 and a new figure legend describing these passage bias sites and epistatic sites.
Point 13: Table 3. The data are interesting. However, the authors should try to find alternative and/or complementary modes of representation, including graphical representation. A number of residues seem to be associated in clusters, both in the MDCK vs EGG and ORI vs EGG for the H3N2 viruses, e.g. [19, 160, 175-176, 210], [154, 158]…, and residue 176 seems to play a prominent role, given the number of times it shows covariations with other residues (with very low P-values). It would probably be useful to locate these associated residues in the HA structure (notably the [19, 160, 175-176, 210] cluster). It is striking that the prominent residues in the co-variation analysis do not appear in the dominant passage bias sites (Table 2). Could the authors propose hypotheses to explain that?
Response 13: We thank the reviewer for the helpful comment and good insight in identifying residue 176 (174 in H1pdm09 absolute numbering). We have now included the epistatic sites along with the passage bias sites in structure Figure 3. Residues 158, 173, 208 and 239 (H1pdm09 absolute numbering) are shared between the passage bias sites and the epistatic sites. As can be seen from Figure 3, majority of the residues undergoing epistatic interactions are in close proximity with the dominant passage bias sites (residues 127, 141 152, 156, 174 in H1pdm09 absolute numbering). We now also included the H1pdm09 absolute numbering in Table 3 to allow easy cross referencing. Hence, majority of these epistatic sites that are influenced by passage bias sites are either overlapping with the most dominant passage bias sites or are in close structural proximity to them.
Point 14: Section 3.3 (lines 156-187). As stated above (comments to Table 1), the approach may be inadequate for the H5 viruses, essentially because of the low number of MDCK passages. The authors are elaborating too much on unsufficient data. Because of the very low fraction of H5 viruses with MDCK passages (47 sequences for a total of 1596 H5 sequences), one may question the statistical validity of the whole analysis on H5 viruses.
Response 14: We agree with the reviewer that the low number of MDCK passages for H5 viruses is problematic if we were to use it to interpret the direction of host adaptations. As mentioned earlier, we have made substantial changes to the analysis looking at subtypes separately and new combinations of the datasets without H5 with a new emphasis on showing that also when excluding H5 data we can use passage data of seasonal viruses (H1p, H1s, H3) to identify known host specificity sites.
Point 15: Lines 193-95. The authors have not really shown that “passage adaptation sites … are strongly associated with … host specificity”. Or at least this endeavour was not really convincing for H5 viruses. Perhaps should they instead try to find corroborating data with H1 or H3 viruses.
Response 15: We agree that this point could be strengthened. In addition to studying the effect of the suggested separation of H5 from the seasonal H1s, H1p and H3, we expanded the comparison to compare two separate time periods to show robustness of the findings as well as an additional subset of cases where the same sample/strain has been passaged with different cell types and show that all of these orthogonal sets agree well with the known sites even for H5 by showing statistically significant overlap with the WHO H5 genetic inventory.
Point 16: The discussion could try to establish useful correlations and to go deeper in the structural relationships between the dominant residues.
Response 16: Same answer as point 2. New figures done.
Point 17: Ref 22 is incomplete.
Response 17: fixed (URL). “H5N1 Genetic Changes Inventory. Available online: https://www.cdc.gov/flu/avianflu/h5n1/inventory.htm (accessed on 28 May 2019).”
Round 2
Reviewer 2 Report
Sorry, I am afraid that I am not persuaded.
Reviewer 3 Report
The authors have brought substantial modifications to their manuscript and have correctly answered the points that were raised. However, there are a few points that need to be addressed (see below).Major remarksLines 130-31 and suppl table 2. I suppose the authors mean “calculating the geometric mean of their odds ratios” (not the geometric mean of the number of different subtypes and different passage cell types)Suppl. table 2 and Fig S2. The comparison of these data is somewhat confusing. There is no legend to suppl. Fig 2, and with no information one would expect that the values (the y values) in the two graphs of suppl. Fig 2 are calculated in the same manner as the last column of suppl. table 2. (even if the geometric mean of the odds ratio is calculated separately for each subtype in the graphs, the result should be somewhat similar to that in suppl. table 2).Table 2 and lines 142-144. There are 54 dominant sites listed, but it is unclear how these 54 sites were arrived at. Is it 31+19+4 (Fig 2C) or 19+26+4+5 (Fig 2D)? In either case, the 19 residues shared by the three methods should be highlighted in table 2. And the 4th column in Table 2 is also somewhat unclear. The 5th colum is clear, since it corresponds exactly to the last column of suppl. table2. However, to increase the readability and to make the correspondence even clearer, I suggest to highlight the corresponding lines in suppl. table 2 in different colours.Further, it is not perfectly clear why table 2 contains some residues with very modest scores (below 5) but lacks some with high scores (for example, 169 (H3 numbering), with a score of 11.88). This is perhaps after taking into account the SSDPs, but it is not clearly explained.Minor remarksLine 78. ..some viral strains…Line 204 …. to give rise…Author Response
Please see the attachment.
